# Balancing Equity and Efficiency in the Allocation of Health Resources—Where Is the Middle Ground?

**DOI:** 10.3390/healthcare9101257

**Published:** 2021-09-24

**Authors:** James Avoka Asamani, Samuel Anongiba Alugsi, Hamza Ismaila, Juliet Nabyonga-Orem

**Affiliations:** 1Universal Health Coverage-Life Course Cluster Inter-Country Support Team for Eastern and Southern Africa, Regional Office for Africa, World Health Organisation, 82–86 Cnr Enterprise/Glenara Roads, Harare CY 348, Zimbabwe; nabyongaj@who.int; 2Faculty of Health and Social Sciences, Bournemouth University, Bournemouth BH8 8AJ, UK; salugsi@bournemouth.ac.uk; 3Ghana Health Service-Headquarters, Private Mail Bag, Ministries, Accra 23302, Ghana; hamza.ismaila@ghsmail.org; 4Centre for Health Professions Education, Faculty of Health Sciences, North-West University, Potchefstroom Campus, Building PC-G16, Office 101,11 Hoffman St., Potchefstroom 2520, South Africa

**Keywords:** equity, efficiency, resource allocation, healthcare

## Abstract

The notion of equity in health service delivery has been embodied in several of the Global Sustainable Development Goals (SDGs), especially the aspiration for universal health coverage (UHC). At the same time, escalating healthcare costs amidst dwindling resources continue to ignite discussions on the efficiency aspect of healthcare delivery at both operational and system levels. Therefore, health planners and managers have had to grapple with balancing the two, given limited resources and sophisticated population health needs. Undoubtedly, the concepts of equity and efficiency have overarching importance in healthcare. While efficiency dictates an ‘economical’ use of the limited healthcare resources, equity advocates their fair and ethical use. Some have leaned on this to argue that one has to be forgone in search of the other. In search of a ‘middle ground’, this paper explores the conceptual underpinnings of equity and efficiency in the context of healthcare resource allocation with some empirical examples from high-income and low- and middle-income settings. We conclude by arguing that equity and efficiency are, and ought to be, treated as complementary rather than conflicting considerations in distributing health resources. Each could be pursued without necessarily compromising the other—what matters is an explicit criterion of what will be ‘equitable’ in ensuring efficient allocation of resources, and on the other hand, what options will be considered more ‘efficient’ when equity objectives are pursued. Thus, equity can be achieved in an efficient way, while efficiency can drive the attainment of equity.

## 1. Introduction

Governments and policymakers seek to meet the health needs of their citizens. However, the need for health and healthcare are unending, and resources are not limitless. Therefore, the resource allocation decision must always be made on the basis of equity and efficiency. While the principle of efficiency seeks to maximise the total population health given the resource constraints, the notion of equity concerns fairness in distributing health and healthcare aimed at minimising any differences amongst population groups [1,2]. It has, therefore, been argued that health systems across the world have two broad objectives—efficiency and equity, which tend to have a counter effect on each other [3]. Policymakers would then have to find an appropriate balance in the pursuit of these objectives.

The aforementioned has generated an ongoing debate in the health economics literature. While many authorities have proposed methods of making trade-offs between the two under a given circumstance [1,4,5], the fundamental question has been whether equity and efficiency indeed conflict or rather complement each other.

This paper examines the theoretical arguments of equity and efficiency in healthcare and explores empirical work in the published literature. It concludes by arguing that equity and efficiency are, and ought to be, treated as complementary rather than conflicting considerations in distributing health resources.

## 2. Theoretical Concepts on Equity in Health Resource Distribution

Equity has often been defined in terms of fairness and justice, or the notion of balancing reasonable, competing claims of people in a society in an unbiased and objective manner [6]. In general, normative economic analysts and policymakers have often discussed equity in healthcare in terms of fair distribution of some ‘good’. However, unless the particular good of interest is explicitly defined, equity could mean different things to different people or the same people under different circumstances. Therefore, the question of ‘equity of what?’ must be core to any discussion around equity [7]. Sen [8] termed the variable for distribution as the ‘focal variable’. Given the diversity of human needs, attaining equality in one or more focal variables tends to create or deepen inequalities in other competing focal variables [6,9]. It has been argued that the focal variable for distribution should be healthcare [10]. However, human demand for healthcare is derived from the need for health itself [11]; therefore, others have asserted that instead of trying to equitably distribute healthcare, the focus should be the equitable distribution of health [12]. Another school of thought believes that some healthcare (or products such as cosmetic surgery, sex enhancement drugs) are consumed for their direct utility [13], and thus, utility should be the focal variable of distribution. It would appear that the latter argument is in line with the welfare economic stance, whereas those who favour distributing health tend to lean towards extra-welfarism [13,14].

### 2.1. Some Equity Concepts Explained

The health economics literature is replete with different normative concepts of equity, although only a few have been empirically tested and have accompanying tools to measure (in)equity. Policymakers, therefore, need to be explicit on their normative stance on the equity concept for distributing healthcare resources. Some of these concepts and their implications for health policy are explored herein.

First, utilitarianism is one equity principle based on the ethical standpoint of achieving the greatest good for the greatest number of people [2]. This implies that policymakers should concentrate on what would benefit more people even if it is marginal, as opposed to enormous improvements for a minority. However, this could breed inequity. For instance, the social health insurance policy in Ghana excluded the treatment of some diseases deemed to be affecting only a few people in the country (for example, dialysis for chronic kidney disease) [15]. It is often the case that despite their real need, those afflicted cannot afford the treatment and risk dying, a situation creating or deepening financial inequity in access to health services.

Another equity consideration is equality of health. This stance prescribes that health, measured in some form (such as quality-adjusted life years (QALYs)), ought to be distributed equally among all persons [16]. Culyer [14] argues that this could imply equal distribution of all the inputs required for the attainment of health, such as healthcare, food, housing, and potable water, etc. However, many of these determinants are often outside the control of the health sector, while lifestyle choices such as smoking and alcohol consumption may also produce utility for the consumers. Thus, an attempt to control people’s lifestyles for the sake of achieving equality in health may conflict with their freedom of choice. Furthermore, equal distribution of the inputs or a certain permutation of the same may not necessarily yield the same amount of health in different population groups. Nonetheless, health is the basis for human flourishing, and therefore ensuring that it is distributed equitably is an ethical imperative [7,17]—just that how to go about distributing health itself presents an intriguing puzzle.

Moreover, equality of need, the idea that healthcare resources be distributed according to ‘need’ has a strong instinctive appeal and appears to be pursued by most policymakers [18]. However, what is deemed as ‘need’ is defined differently in different contexts [7], which tend to impact the judgement of whether an allocation is equitable or not. One of such definitions equates healthcare needs to the degree of ill health such that it is those most severely ill who would be deemed to have the greatest need. In the context of distributing health resources to regions or districts in a country, this stance would imply that the regions or districts with poor health outcomes receive the most allocation. However, this could also lead to a moral hazard where poor health outcomes become desirable (and deliberately pursued) in anticipation of a reward with greater resource allocation [17].

Health economists also define need in terms of ‘capacity to benefit’ from intervention and not merely the severity of ill health [19]. It follows, therefore, that if the persons most in need of healthcare are also those capable of benefiting from it, then the objective of maximising health gain (efficiency) is consonant with fairness (equity), with no ethical or technical contradictions. Thus, the same distribution of resources attains both efficiency and equity [3].

Equality of access is also one equity standpoint advocating the allocation of healthcare resources in a manner that all persons have equal access to healthcare [2]. Culyer and Newhouse [6] define equality of access to healthcare as every individual being equally able to obtain or make use of healthcare. It would seem that this relates only to the ability or capacity to obtain healthcare and not necessarily whether it is actually obtained. The service availability is one thing, and the individuals’ decision or preference of using it is another. For this reason, Mooney [17] points out that equity in healthcare access should not necessarily be evaluated on the basis of utilisation patterns. Bevan et al. [20] contend that it is founded on the notion of equal opportunity to receiving the care and not about its eventual usage. Empirically, Johns et al. [21] found that people in Afghanistan were more concerned about the provision of equal opportunities to orthodox treatment so that the decision to use it or resort to traditional treatment is left to them to choose.

It seems as though making allocations based on the principle of equality of access would be ethically more appealing when considered alongside the concept of need. Thus, the idea of equal access for equal need could address both equity and efficiency concerns [5,14,18].

Others have also held the view that resource distribution in health ought to consider equality of expenditure. By this, each individual is deemed to be entitled to an equal share of the total healthcare expenditure [17]. This stance informs economic analysis using measures of healthcare expenditure per capita [7]. This has, however, been criticised for ignoring the fact that people access healthcare largely based on need and preference, not necessarily having the money.

Another ethical argument advanced by Alan Williams is the concept of fair innings [22]. The proponents of this notion assert that everyone is entitled to a certain lifespan of some measure (e.g., QALYs) and that those who obtain less are ‘cheated’, whereas those who obtain more live on ‘borrowed time’. In resource distribution, this implies that more resources should be given for the care of those who have not had their fair innings; for example, more weight to be given to childcare, as compared to geriatric care. This, according to Williams and Cookson [23], would bridge intergenerational inequalities. Indeed, some empirical literature has reported that policymakers prioritise the care of younger people, as compared to older persons [1,5].

Although the aforementioned concepts are far from an exhaustive discussion of all the normative stances on equity, the central point is that different concepts of equity tend to have substantially different outcomes regarding what would be considered a fair and just distribution of health and healthcare resources [14].

### 2.2. Horizontal and Vertical Equity

Even though several equity concepts have been espoused in the literature, horizontal and vertical equity have often been empirically tested and featured in most equity discussions [13].

Horizontal equity is defined in terms of ‘equal treatment of equals’ [6]. Thus, those who have similar situations in terms of the focal variable should be allocated similar resources. In terms of health financing, this would mean that those with similar income levels contribute similarly towards healthcare or that those with similar health needs are given similar treatment irrespective of their ability to pay, geographical location, or any other non-health variable [3,4,24]. The English National Health Service (NHS) was founded on this fundamental principle of horizontal equity [25].

On the other hand, vertical equity dictates ‘unequal treatment of unequals’ [6,17]. That is, those who have different situations in terms of the focal variable are treated differently. Therefore, unequal treatment should be given to unequal people in proportion to the extent to which they are unequal [6]. Healthcare funding based on this principle would demand that persons who earn higher income pay greater tax or insurance premium than a person who earns lower income, and also greater resources should be given to those who have greater health needs. Simply put, there can be ‘equitable inequalities’ synonymous with the Marxist notion ‘*…from each according to their ability; and to each according to their need’* [14]. Achieving vertical equity largely requires some form of ‘positive discrimination’ or ‘deliberate unequal distribution’ [22]. An important implication of the distinction between horizontal and vertical equity is that what is regarded as equitable often entails inequality in health and inequality in resource distribution.

Several tools have been developed to measure equity (or inequity), including concentration index, Gini coefficient, Lorenz curve, Theil index, Atkinson index, index of dissimilarity, etc. [26]. These are beyond the scope of this paper.

## 3. Theoretical Concepts on Efficiency in Healthcare

In the context of health, efficiency is often used to mean the maximisation of population health, given a certain amount of resources [1]. Mechanistically, efficiency is a measure of the extent to which inputs are put to good use to achieve the intended function or output [27]. This implies that efficiency is relative productivity, given that productivity is deemed as a ratio of resource consumption (inputs) to results (outputs) [28]. Therefore, efficiency can simply be seen as increasing output with constant or reduced inputs or achieving the same output with minimal input [3].

In general, economists attempt to distinguish three interrelated levels of efficiency: technical, cost-effective, and allocative. When the production function is organised to achieve a given output with minimum input, technical efficiency is attained [26], for instance, when relatively inexpensive drugs rather than surgery are used to treat uncomplicated appendicitis to achieve the same speed and level of recovery.

Relatedly, when the production is organised with the aim of achieving a given output with minimal cost, cost-effective efficiency is said to be pursued [6]. This means the focus is on cost minimisation without compromise on output. Therefore, this depends on both the production function and the prevailing cost of inputs. For instance, Ciprolex-TZ (a single tablet containing ciprofloxacin and tinidazole) is known to be effective against many intestinal infections. However, in many contexts, it is expensive and must be prescribed by only qualified medical doctors. Alternatively, septrin and metronidazole taken as separate tablets can achieve the same results. These are not only cheaper alternatives but could be prescribed by nurse practitioners and physician assistants who also represent a cheaper labour input [29,30]. Indeed, in aggregate health system efficiency analysis, studies have shown that, on the one hand, human resources for health and medicines availability are critical inputs and, on the other hand, they have been implicated as the drivers of inefficiency [31,32]. Therefore, taking these factors into account in an empirical analysis of efficiency is imperative.

From the foregoing, it would be observed that technical efficiency is a prerequisite for cost-effectiveness, both of which are supply-side concepts of efficiency because the consumer is not considered in these concepts. Cost-effectiveness analysis (CEA), as the name suggests, is one of the health economic tools that address these types of efficiency.

On the other hand, allocative efficiency incorporates the demand side by addressing the ‘optimal’ production of resources and the distribution of the same to consumers in line with the value placed on them or according to consumer need [6,23]. Thus, beyond being technically efficient and cost effective, health/healthcare also needs to be offered to the persons in need of it. Therefore, one would argue that allocative efficiency embodies the maximisation of total health at the lowest cost and distributing the same to those who need it. Thus, it incorporates equity and cost-containment concerns. Cost–utility analysis is what is the main tool that addresses allocative efficiency, though other techniques such as programme-based budgeting and marginal analysis (PBMA) and multiple-criterion decision analysis (MCDA) also attempt to achieve this.

Hurley [6] observes that there is a hierarchical relationship between the efficiency concepts such that technical efficiency is a necessary condition for cost-effectiveness, both of which are necessary conditions for allocative efficiency. Culyer [14] extends this argument by pointing out that ensuring equity in the distribution of health resources must be carried out within the context of maximising the gains (efficiency). Thus, allocative efficiency is a necessary condition for equity and fairness [3]. From the foregoing, it would seem that equity and efficiency objectives can potentially complement rather than conflict with each other.

## 4. Empirical Examples of Equity and Efficiency Considerations in Healthcare

Even though a number of tools exist for assessing equity and efficiency of health systems [13,26,33], there appears to be a paucity of studies that examined both concepts within or across health systems. Studies have focused on either measuring equity, efficiency or eliciting decision maker’s views of a trade-off between the two [1,5,21,27,34]. Nonetheless, the findings of these studies have significant implications for discussing the contradictions or otherwise between equity and efficiency in resource allocation.

In the context of a developed country, Kontodimopoulos et al. [35] used data envelopment analysis (DEA) to examine the efficiency of hospital health centres in Greece. These are hospitals located in remote areas and serve relatively small local populations with the primary objective of providing equity in terms of access to healthcare. The researchers concluded that 25.13–26.77% resource use inefficiencies were apparent in these hospitals, as compared to their urban counterparts. Thus, up to a quarter of the healthcare resource is traded off in order to achieve equity in access to basic healthcare in these populations. Indeed, similar trade-offs between equity and efficiency objectives have been documented in the UK [34] and other European countries [20,24].

From developing countries, Johns et al. [21] concluded that in Afghanistan, remote health facilities were under-utilised by 13% when compared with those in relatively urban areas, but that the quality of care was comparable. They further contended that those in remote areas were ensuring horizontal equity at an average labour cost of USD 1.44 per patient visit, compared with USD 0.97 at urban facilities. This would seem to suggest that achieving horizontal equity required at least a 32.6% trade-off in efficiency. On the contrary, Alhassan et al. [27] reported that health facilities located in rural parts of western Ghana had higher odds (42.9–52.1%) of being efficient, as compared to those in the capital city (*p* = 0.002). In a recent study to evaluate the trade-offs between geospatial equity and health systems efficiency in Kenya, Malawi, Tanzania, and Rwanda, Iyer et al. [36] found evidence to suggest that in prioritising efficiency, considerate choices with regard to geographical allocation of health resources could enhance equitable physical access to services.

In addition, whilst the findings of some studies such as those by Johns et al. and Kontodimopoulos et al. [21,35] appear instructive, one would also argue that if healthcare provision in urban areas were organised to minimise the input cost, the efficiency losses in the rural areas would be offset by the efficiency gains made at the urban areas [3]. Thus, the principle of maximising health for the citizenry would be attained, and those requiring it would indeed receive the same, in which case the distribution would be equitable and with some allocative efficiency.

In assessing resource allocation priorities, Jehu-Appiah et al. [1] used MCDA to explore the practices of policymakers in Ghana. The study reports that by using consensus building, policymakers consider vulnerable populations alongside cost-effectiveness as the basis for resource allocation. Similarly, the World Bank [37] reported that Chinese policymakers incorporate efficiency and equity in their decision making by using league tables. In this Chinese context, Paolucci et al. [5] sought to test Culyer’s claim that there is no conflict between efficiency and equity using an MCDA. The authors, in supporting Culyer’s argument, concluded that China’s approach of complementarily addressing both equity and efficiency concerns has reduced health disparities and kept healthcare costs relatively low. A recent analysis of Croatia’s health system suggests that it is one of the most efficient health systems in Europe but that inequities in relation to unmet health needs are wide. Thus, much was still needed to be carried out to translate the efficiency gains into bridging the health inequalities [38]. This means that a solo pursuit of efficiency without equity considerations could hurt the overall health system objectives.

From the foregoing, it would seem reasonable to argue that reducing disparities in health and healthcare (or equity) has a significant but intricate link with maximising universal health outcomes (or efficiency). As Culyer sums it, ‘…an inefficient allocation can be equitable; an efficient allocation can be inequitable; an inefficient allocation can become more efficient without increasing inequity; [and] an inequitable allocation can become more equitable without reducing efficiency’ [14]. Essentially, equity and efficiency do not necessarily contradict; what is deemed equitable in large part also depends on the equity criteria selected, and therefore, one needs to make their stand explicit. However, the complementary effects of equity and efficiency can be harnessed for the greater benefit of society through a deliberative process of resource allocation [1,5,18]. An empirical analysis in the context of New Zealand concluded that there are multiple pathways of maximising efficiency while achieving equity and that, ‘trade-off between equity and efficiency does not always exist’, advancing the argument that improvements in efficiency (up to its maximum) can also yield gains in equity via reductions in inequalities in life expectancy of the population [39].

## 5. Conclusions

There is a wealth of literature addressing equity and efficiency trade-offs which tend to suggest that a contradiction exists between the two concepts. Whilst equity concerns the just and fair distribution of resources, it states nothing about whether the output is produced maximally given the inputs—efficiency. Additionally, merely concentrating on maximising health/healthcare does not necessarily address the issue of whether those who need it indeed obtain it in the right quantity and quality without undue hardship.

It is therefore argued that it is unwarranted and potentially counter-productive to pursue equity and efficiency as separate health objectives. Consequently, Culyer and Bombard [18] proposed a deliberative framework for incorporating both equity and efficiency as complementary concepts in the pursuit of maximal and equal health for populations. In the Sustainable Development Goals, especially Goal 3, ensuring universal health coverage is a global priority within which equity in access to services and financial protection are the main tenets. However, in the face of escalating health expenditure, improving access to services to those who lack it and ensuring protection from catastrophic expenditures for those who need it are not just questions of increasing health expenditure but how well available resources are used. Hence, more than ever, making efficiency gains from which to pursue equity objectives is imperative if UHC is to be attained.

In conclusion, we contend that what is equitable depends on the equity criteria selected; equity and efficiency do not inherently contradict. Understandably, achieving the objectives of equity may not always be cost neutral and can require additional resources, but this can be offset by efficiency gains from other aspects of the health system. In the long run, the dividends of achieving equity objectives such as improved health of the population will contribute to production efficiency in the economy. Although Culyer put forward a similar argument severally, more than ever, in the pursuit of universal health coverage, efficiency must work for equity, and equity, in turn, improves efficiency in the long run. Thus, equity and efficiency are, and ought to be, complementary for the attainment of the health objectives of any country.

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
