# Peer review of "Balancing Equity and Efficiency in the Allocation of Health Resources—Where Is the Middle Ground?"

_healthcare, 2021, doi:10.3390/healthcare9101257_

Round 1

Reviewer 1 Report

Thank you for giving me the opportunity to review the perspective article. The author discussed on balancing equity and efficiency in the allocation of health resources. The topic is socially important, but the there are several problems in the manuscript. Therefore, the reviewer thought that the manuscript should be revised before further considerations. I left comments below.

Comments:

Authors’ Affiliation:

  1. The authors should write the details of their affiliations.

Theoretical Concepts on Equity in Health Resource Distribution:

  1. The authors should explain the theoretical concepts using (a) figure(s). It should be informative for potential readers.

2.1. Some equity concepts explained:

  1. The authors explained several equity concepts. Even if the name of the concept (utilitarianism, equity of health, equity of needs, and othres) is the same, the implications may vary from country to country. The authors should explain about it in detail with relevant references.

Theoretical Concepts on Efficiency in Healthcare:

  1. Can the “efficiency” be explained only by the aspects which presented in this section? The reviewer understood the supply-side and the consumer-side concepts, but it would also be necessary to mention the deployment of human capital.
  2. The authors should also discuss the medical equipment (medical environment) aspect.

Empirical Examples of Equity and Efficiency Considerations in Healthcare:

  1. The authors should add other empirical examples. Only limited examples included in the current manuscript.
  2. The authors should explain the reason why they select the examples in the manuscript.

Author Response

We are grateful for your constructive feedback which has guided us to revise the manuscript. We hope that the manuscript is greatly improved based on your kind feedback. We have attached the revised manuscript and a point-by-point response to the review comments for your consideration.

Reviewer 2 Report

I enjoyed reading this article. It is well written and addresses an interesting topic. I have three broad comments.

First, this article is building an argument based on what is known, and previously argued, in the existing literature. However, the majority of the work cited is relatively old (most paper pre-date 2015). Are there any recite contributions to this argument?

Second, several definitions of equality are proposed in Section 2. However, it is not clear which definition is used for the conclusion relating to the complimentary of equity and efficiency. This is important for your conclusion and should be very clear to the reader.   

Third, the contribution of this article, particularly over Culyer (2015), needs clarification.  In my reading of the article, the conclusion agrees with Culyer (2015), but does not have anything to add over and above it.

Author Response

(The authors gave the same response as above.)

Round 2

Reviewer 1 Report

Thank you for giving me the opportunity to review the revised version of this article. The author corrected the manuscript according to the comments. Therefore, the reviewer thought that the manuscript can be accepted for publication in the journal.